# Valorization of Lignin and Its Derivatives Using Yeast

Filemon Jalu Nusantara Putra [1], Prihardi Kahar [1,*], Akihiko Kondo [1,2] and Chiaki Ogino [1,*]

1 Department of Chemical Science and Engineering, Graduate School of Engineering, Kobe University, 1-1 Rokkodaicho, Nada-ku, Kobe 657-8501, Japan

2 Graduate School of Science, Technology and Innovation, Kobe University, 1-1 Rokkodaicho, Nada-ku, Kobe 657-8501, Japan

* Correspondence: pri@port.kobe-u.ac.jp (P.K.); ochiaki@port.kobe-u.ac.jp (C.O.)

**Abstract:** As the third most plentiful biopolymer after other lignocellulosic derivates such as cellulose and hemicellulose, lignin carries abundant potential as a substitute for petroleum-based products. However, the efficient, practical, value-added product valorization of lignin remains quite challenging. Although several studies have reviewed the valorization of lignin by microorganisms, this present review covers recent studies on the valorization of lignin by employing yeast to obtain products such as single-cell oils (SCOs), enzymes, and other chemical compounds. The use of yeasts has been found to be suitable for the biological conversion of lignin and might provide new insights for future research to develop a yeast strain for lignin to produce other valuable chemical compounds.

**Keywords:** yeast; lignin; single cell oil; enzymes; value-added products

## 1. Introduction

The depletion of fossil fuels has given rise to concerns in recent years [1]. As the most plentiful aromatic biopolymer on the earth, representing around 300 billion metric tons [2], lignin offers potential for biofuels and diverse chemical production by means of biorefinery processes [3]. It is expected to profit the global economy while maintaining sustainable development [4]. However, currently, most lignin substances are decomposed by burning [5]. Achieving success in sustainable lignin valorization has been quite challenging so far [6]. The main reasons for this are mostly due to its rigid, complex structures, strong smell, and its toxicity to some living organisms [7]. Certain biochemical approaches, like thermochemical combined with enzymatic techniques, are commonly applied to gain value-added lignin compounds [8]. Research related to the valorization of lignin via the use of microorganisms has gained interest because of the potential to develop low-emission processes yielding valuable biofuels and chemical compounds. The success of future biorefineries may depend on novel approaches to lignin utilization. This review highlights the current situation of lignin valorization employing yeasts as a microbial platform to produce valuable products.

### 1.1. Lignin Resources

Lignocellulosic biomass is mainly constituted of cellulose, hemicellulose, and lignin, with the composition varying depending on the type of biomass (Table 1). Lignin is a phenylpropane polymer unit [9] that builds up a strong integrated system with cellulose and hemicellulose in biomass over covalent and hydrogenic linkages [10]. Lignin, as a phenylpropanoid biopolymer, gives rigidity to the biomass structure [11]. In plants, lignin provides important functions against environmental stresses such as diseases from microorganisms and oxidative stress [12]. However, the potential of lignin remains to be explored due to the complexity of its structure compared with cellulose and hemicellulose [13]. Several depolymerization methods, such as physical, chemical, and biological, have been successfully applied to lignin to produce monomers and oligomers [14]. Monomers and

oligomers can be used as substrates for biofuel and various chemicals [15,16]. The global annual production of lignin by the paper and ethanol industries is around 100 million tons [17]. Most lignin waste is used as a low-value substrate to produce heat and generate electricity [18]. The quantity of lignin produced by these industries is predicted to increase in the following years, especially in the biorefinery industry, due to the use of renewable energy products such as fuels and chemicals.

**Table 1.** Relative composition of lignocellulosic biomass.

| Lignocellulosic Biomass | Lignin | Hemicellulose | Cellulose | Reference |
|---|---|---|---|---|
| | (%) | (%) | (%) | |
| Hardwood | | | | |
| Aspen | 19.5 | 21.7 | 52.7 | [19] |
| Beech | 20 | 33 | 45 | [20] |
| Cherry wood | 18 | 29 | 46 | [20] |
| Poplar | 20 | 24 | 49 | [20] |
| Willow | 29.3 | 16.7 | 41.7 | [19] |
| Softwood | | | | |
| Fir | 30 | 22 | 45 | [20] |
| Pine armandii | 24.1 | 17.8 | 48.4 | [21] |
| Japanese cedar | 33.8 | 23.1 | 38.6 | [22] |
| Spruce | 27.6 | 29.4 | 43 | [23] |
| Others | | | | |
| Barley straw | 14–19 | 27–38 | 31–45 | [24] |
| Bamboo | 20.81 | 19.49 | 39.8 | [22] |
| Corn cobs | 18.2 | 33.1 | 34.6 | [22] |
| Corn strover | 7–21 | 28 | 38–40 | [24] |
| Rice straw | 12–14 | 23–28 | 28–36 | [24] |
| Wheat straw | 20.2 | 34.4 | 37.5 | [22] |
| Banana waste | 14 | 14.8 | 13.2 | [25] |
| Nut shells | 30–40 | 25–30 | 25–30 | [25] |
| Coffee grounds | 19.8–26.5 | 5–10 | 59.2–62.9 | [26] |
| Newspaper | 18–30 | 25–40 | 40–55 | [25] |

### 1.2. Structure of Lignin

As a natural macromolecule, the structure of lignin contains methoxy, phenolic hydroxyl, and terminal aldehyde groups [27]. Lignin consists of three types of phenylpropanoids, i.e., *p*-hydroxyphenyl (H), guaiacyl (G), and syringyl (S) [28], composed with *p*-coumaryl alcohol, coniferyl alcohol, sinapyl alcohol, respectively, as monomers (Figure 1a) [6,28,29]. At a glance, the chemical structures of phenylpropanoids appear similar; their differences depend on the interchange degree of the methoxy groups in the aromatic rings [30]. These three monolignols generate phenoxyl radials, randomly polymerized to form biopolymers with three-dimensional networks. The weighted average molecular weights ($M_w$) of isolated lignin (milled wood lignin) from *Eucalyptus globulus*, Southern pine, and Norway spruce are 6700, 14,900, and 23,500 Da, respectively [31], with the molecular weight varying considerably according to the isolation method. The Klason method measures lignin contents in softwood at 25–35%, hardwood at 20–25%, and herbaceous plants at 15–25% [32]. Understanding the phenylpropanoid structures is essential for choosing an appropriate pretreatment method, mainly when lignin dissolves in the solvent [33]. The unit of phenylpropanoids in lignin are attached by chemical bonding, namely α-O-4, β-β, β-1, β-5, β-O-4, 5-5, and 4-O-5 (Figure 1b). The most abundant bond is β-O-4, comprising around 50% of all bonds [34]. Most major bonds have low bond-dissociation energy (BDE) values, making it possible to convert lignin into other compounds [35]. The three phenylpropane building blocks of lignin correspond to the lignin structures *p*-hydroxyphenyl, guaiacyl, and syringyl. Softwood lignin consists primarily of G-units with traces of H-units, whereas hardwood lignin contains both G- and S-units.

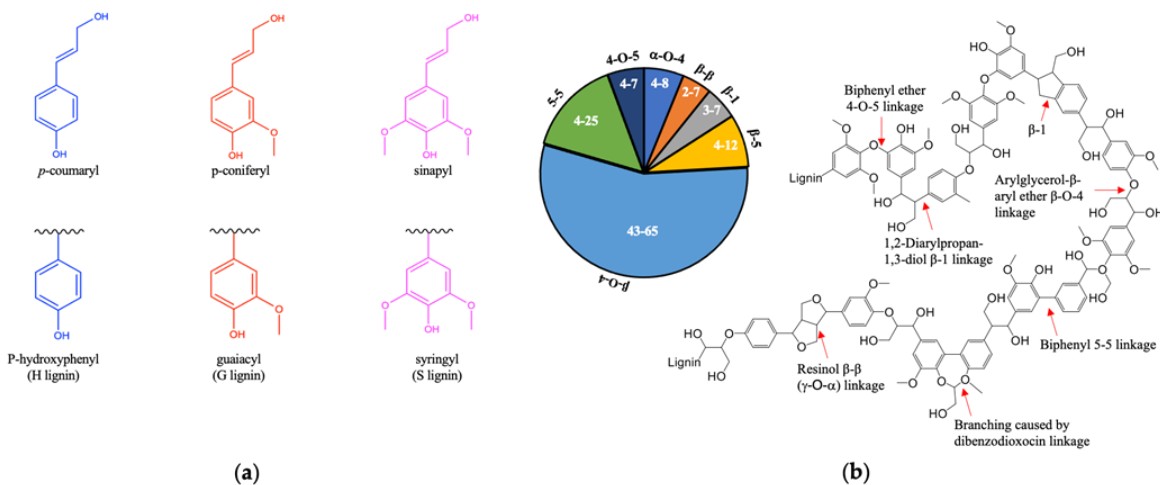

**Figure 1.** Relative structures of three monomeric subunits in lignin (**a**). Linkage amount and its position in lignin structure (number/100 phenylpropane units) (**b**). These are modified versions from previous reports [28,29].

## 2. Current Lignin Valorization by Microbes

In the last decade, microbes have produced biofuels and high-value products such as lipids, vanillin, pyrogallol, *cis*, *cis*-mucoid acid, lactate, succinate, ferulate, pyridine, and biopolymers (PHAs, monolignols) from lignin and its derivates [36–41]. The breakdown of the lignin is the initial step in synthesizing biofuels and biomaterials. To this end, enzymatic degradation, specifically by microorganisms, is critical to obtain the compounds for lignin utilization and may become a promising eco-friendly method in the near future. All lignin-degrading enzymes, such as laccases, lignin peroxidases, manganese peroxidases, versatile peroxidases, and dye-decolorizing peroxidases, can be produced by microorganisms such as fungi, bacteria, and yeasts [29]. In recent years, the development of strains and their application as lignin-degrading enzymes have been reported [42].

## 3. The Use of Yeasts

### 3.1. Lipids

Many studies have focused on lipid production, i.e., single cell oils (SCOs), by microorganisms [43]. SCOs are positioned as the third generation of biofuels, and research has been reported based on the use of several microorganisms, such as, bacteria, microalgae, fungi, and yeast, that can utilize lipids [44]. Most model microorganisms that have been reported to accumulate lipids from lignin and its derivates are microbes. Compelling research has characterized lignin metabolism by *Rhodococcus opacus* and *R. rhodochrous* [45,46]. Notably, microbial metabolism, especially that of yeast, has merits thanks to its ability to deal with several aromatic compounds due to its cell tolerance [47].

The combination of the metabolism of aromatic compounds to fatty acids may lead to the possible large/industrial-scale production of biofuels and other chemicals from lignin and its derivatives in the future [48]. *Trichosporon* has gained attention in the past few years due to its ability to produce large quantities of lipids using lignocellulosic substrates and high tolerance to various lignocellulosic inhibitors [49–52]. The potential ability of *Trichosporon* to grow on lignin derivatives was studied using *T. cutaneum*. *T. cutaneum* ACCC 20271 yeast reportedly grows on lignin-derived phenolic aldehydes such as 4-hydroxybenzaldehyde, vanillin, and syringaldehyde. Research has shown that *T. cutaneum* has better tolerance to 4-hydroxybenzaldehyde (1.5 g/L) compared with vanillin (0.1 g/L) and syringaldehyde (0.5 g/L). In the fermentation process, 4-hydroxybenzaldehyde, as a representative of *p*-hydroxyphenyl or H lignin, was found to be suitable for use as a substrate for *T. cutaneum* for lipid production (16.6% or 0.85 g/L). At the same time, the other phenolic aldehydes (vanillin and syringaldehyde) were converted

to alcohols and acids. Then, 4-hydroxybenzaldehyde is converted into 4-hydroxybenzoate by aldehyde dehydrogenases of *T. cutaneum.* Afterward, it is assimilated into protocatechuate by dioxygenases with oxygen as the substrate. Protocatechuate is converted into acetyl-CoA and succinyl-CoA via the β-ketoadipate pathway. Acetyl-CoA is used for lipid synthesis directly or is assimilated first into the TCA cycle and used as the dominant precursor for lipid synthesis [53]. Subsequently, *T. cutaneum* ACCC 20271 and MP11 stains demonstrated the tolerance of 4-hydroxybenzaldehyde (0.8 g/L), 4-hydroxy-3-methoxybenzaldehyde (0.8 g/L), and syringaldehyde (0.6 g/L) in wheat straw hydrolysate by biodegrading these aldehydes with adequate cell growth and lipid accumulation (40.87%) [54]. Another species from *Trichosporon,* named *T. oleaginosus,* has been reported to accumulate lipids on several aromatic compounds (4-hydroxybenzoic acid, phenol, and resorcinol). *T. oleaginosus* ATCC 20509 could endure and metabolize aromatic substrates by ortho-cleavage aromatic metabolism pathways. Afterward, the fed-batch feeding strategy of *T. oleaginosus* in resorcinol revealed a lipid production of 69.5% (1.64 g/L) [55].

Several oleaginous yeast species have demonstrated the ability to metabolize lignin aromatic compounds. The oleaginous red yeast *Rhodotorula toruloides* can metabolize *p*-coumaric acid, ferulic acid, vanillic acid, and 4-hydroxybenzoic acid [56]. *Lipomyces starkeyi* has received significant attention as a SCOs producer due to its ability to accumulate lipids at quantities of up to 70% in a nitrogen-limited mineral medium. It also showed the ability to reduce lignin derivates such as syringaldehyde and vanillin [57]. Another *Cutaneotrichosporon* genus, i.e., *C. guehoae*, reportedly utilized 4-hydroxybenzoic acid as the sole carbon source [58]. However, the studies above did not clearly describe the correlation of lipid production by the respective yeasts.

### 3.2. Enzyme for Lignin Degradation

Enzymes are essential to catalyzing lignin into their derivatives or aromatic monomers, which provide for building blocks of some valuable chemical products. The production of suitable enzymes is currently being explored [59]. Lignin-degrading enzymes were discovered around a hundred years ago in Basidiomycota fungus *Phanerochaete chrysporium* for peroxidases and laccases in a plant named *Rhus vernicifera* [60]. Later, several enzymes that can depolymerize lignin were found in fungi and bacteria [61,62]. Lignin-degrading enzymes are divided into two groups (Figure 2): laccases and peroxidases [63]. These enzymes are used in vitro applications to depolymerize lignin and its derivatives. Implementing in vitro enzymatic conversion has some merits over directed-cell lignin conversion, e.g., it could increase substrate and enzyme interaction while reducing the cultivation time and improving ATP/NAD(P)H imbalance [64]. Nonetheless, yeast plays another role in promising lignin-degrading enzyme producers. Several yeasts, such as *Saccharomyces cerevisiae, Yarrowia lipolytica, Pichia pastoris, Pichia methalonica, Kluyveromyces lactic, Kluyveromyces maxianus*, and *Cryptococcus* sp., were successfully used for the production of laccases and peroxidases by using exogenous genes, not only from fungus-like Ascomycota and Basidiomycota division, but also plants like oomycote and bacteria [65]. The advantages of using yeast for enzyme production include the easy handling of cells, economical substrate for cultivation, rapid growth, quick genetic manipulation, and cell capability for post-translation modification proteins through glycosylation, proteolytic or disulfide mechanisms [66–68].

**Figure 2.** Classification of lignin-degrading enzymes.

### 3.2.1. Laccase Producing Yeasts

Laccases are among the important ligninolytic enzymes produced by microorganisms for lignin depolymerization in lignocellulosic biomass [69]. Lignin degradation by laccase is accomplished by oxidation or receiving an oxygen molecule as an electron acceptor [70]. Yeasts have been shown to produce their laccases via the exogenous laccase gene YlLac [71,72]. Yeast *Rhodotorula mucilaginosa* reportedly produced laccase (3.27 U/mL) naturally due to the presence of the laccase-containing domain 2 (LACC2) gene [73]. In 2017, two yeasts, *Kluyveromyces dobzhanskii* Dw1 and *Pichia manshurica* Dw2, were reported to produce laccases naturally using lignocellulosic biomass as a carbon source. A rice bran substrate supported the laccase production of 88.625 U/L in *K. dobzhanskii* DW1 and sugarcane bagasse of 79.107 U/L in *P. manshurica* DW2 [74]. Yeast, an industrial engineering platform of laccases, shows different potential according to the strain. *Pichia pastoris* can be used for most laccases as expression hosts, such as laccase containing domain 1 (LCC1), LCC2, laccase (LAC), laccase 2 (LAC2), laccase 3 (LAC3), laccase 4 (LAC4), laccase A (LACA), laccase B (LACB), laccase C (LACC) and laccase D (LACD), as well as some recombinant laccases such as LCCA, LCCB, LCCC and LCC5I (Table 2). In the *Kluyveromyces lactis*, the production of laccases POXA1b and POXC from the fungus *Pleurotus ostreatus* yielded higher production than *Saccharomyces cerevisiae* [75]. Laccases produced in yeasts have several applications in diverse fields [76]. One example is the delignification of *Pinus radiata* lignin pulp by Basidiomycota fungi *Coriolopsis gallica* LCC1 laccase produced in *K. lactis* [77]. Additionally, wheat straw lignocellulose was successfully pre-delignified by *P. pastoris* LCC1 laccase from *Pycnoporus sanguineus* H275 [78]. In the implementation of bioremediation, Laccase LCC2 from *Monilinia fructigena* was expressed in *P. pastoris* to reduce the 2, 4, 6-trichlorophenol [79]. Laccase-producing yeasts have also been used for dye decolorizing from fungi such as *Coprinus comatus*, *Coprinus cinereus*, *Lenzites gibbosa*, *Pleurotus sanguineus*, *Trametes genera* like *Trametes trogii* and *Trametes versicolor*, as well as for bacteria including *Thermus thermophilus* [80], *Streptomyces cyaneus* [81]. The resistance present in *S. cerevisiae* was successfully enhanced by expressing the LACA gene from *Trametes* sp. AH28-2 polymerizes 8 mM of vanillin into less toxic compounds [82]. Other researchers have reported the successful development of *S. cerevisiae* in the presence of coniferyl aldehyde by laccases LCC2 of *Trametes versicolor* [83]. The immense laccase activity of yeasts was reported in the successful production of *Cryptococcus* sp. S-2 at 380,000 U/L by LAC2 expression from fungi *Gaeumannomyces graminis* [84]. However, further optimization is needed to use yeast as an industrial host for laccase producers.

### 3.2.2. Peroxidases Producing Yeast

Peroxidases, namely lignin peroxidase (LiP), manganese-dependent peroxidase (MnP), and versatile peroxidase (VP), have immense potential in biotechnology research and industrial applications, as they play an essential role in the production of biofuels and other valuable biochemicals [85]. LiP has similarities with other peroxidases in terms of its catalytic and oxidative systems [86]. LiP, MnP, and VP catalyze lignin-derived compounds, i.e., phenolic and non-phenolic, with hydrogen peroxide ($H_2O_2$) due to the presence of an iron protoporphyrin IX for each mol of protein [87]. Several fungi have been reported to be natural producers of peroxidases such as *Phlebia brevispora* and *P. radiata* from *Phelebia* genera, *Trametes versicolor*, *Phanerochaete chrysosporium*, *Ganoderma lucidum*, and *Coriolopsis occidentalis*. A few bacteria such as *Acinobacter calcoaceticus*, *Klebsiella pneumonia*, and *Streptomyces viridosporus*, reportedly produce peroxidases naturally [63,88]. Meanwhile, a few types of yeast reportedly can produce peroxidase (MnP) are *Meyerozyma carribica* (1884 U/L), *M. guilliermondii* (1779 U/L), *Debaryomyces hansenii* (1806 U/L), and *Vanrija humicola* (1586 U/L) [89,90]. Nonetheless, in the production of peroxidases, yeasts are commonly used as a host for fungal recombinant extracellular enzymes [91]. *P. pastoris* can produce LiP (15 U/L) by expressing the LiPH2 gene from *Phanerochaete chrysosporium* BKM-F-1767 [92]. Later, research on *P. pastoris* and the production of LiP achieved a maximum volumetric activity of 4480 U/L by optimizing the combination strategy in the fed-batch fermentation process [93]. Another *Pichia* genus that can produce LiP by heterologously expressed *Phanerochaete chrysosporium* is *P. methanolica* [94]. Finally, an industrial yeast that can be used as a lignin peroxidases host is *S. cerevisiae*. LiP from *P. chrysosporium* was successfully produced by *S. cerevisiae* and has been used to degrade 2,4-dichlorophenol [95].

In nature, MnP has a critical function in degrading lignin compounds, since it can oxidize $Mn^{2+}$ to $Mn^{3+}$ and diffuse to oxidize lignin and other phenolic compounds [96]. In 2003, MnP was successfully produced in *P. pastoris* by MnP1 from *P. chrysosporium* [97]. The optimization of a recombinant MnP production process based on pH and temperature in a fed-batch fermentation of *P. pastoris* enhanced the enzyme activity (>2000 U/L) [98,99]. Recombinant MnP produced in *P. pastoris* was found in delignification applications to eliminate lignin from hardwood and softwood pulp [100], and MnP from *Ganoderma lucidum* produced in *P. pastoris* was found to be the decolorization platform for four types of dyes, specifically, navy blue HGL, drimaren blue CL-BR, drimaren red K-4Bl, drimaren yellow X-8GN. It also reduced the presence of phenol in the medium [101]. MnP1 from *P. chrysosporium* was also observed in *S. cerevisiae*. It produced manganese peroxidase to increase the growth of cells that was previously inhibited by the presence of toxic compounds [102]. Later, a novel MnP3 gene from white-rot fungi named *Cerrena unicolor* BB6P was successfully expressed in *P. pastoris*, showing 154.5 U/L of MnP activity [103]. Besides that, *S. cerevisiae* was successfully used with other peroxidases such as VP from *Pleurotus eryngii*, i.e., king trumpet mushroom [104].

**Table 2.** Lignin-degrading enzymes produced by yeast.

| Enzymes | Native | Gene | Yeast | Reference |
|---|---|---|---|---|
| Laccase | *Pleurotus ostreatus* | POXA1b | *Kluyveromyces lactis* | [75] |
| | | POXC | *Saccharomyces cerevisiae* | |
| | *Coriolopsis gallic* | LCC1 | *Kluyveromyces lactis* | [77] |
| | *Pycnoporus sanguineus* H275 | LCC1 | *Pichia pastoris* | [78] |
| | *Trametes trogii* | LCC1 | *Pichia pastoris* | [105] |
| | *Trametes trogii* | LCC1 | *Kluyveromyces lactis* | [106] |
| | *Trametes versicolor* | LCC1 | *Pichia methalonica* | [107] |
| | *Monilinia fructigena* | LCC2 | *Pichia pastoris* | [79] |
| | *Trametes versicolor* | LCC2 | *Saccharomyces cerevisiae* | [83] |
| | *Lenzites gibbosa* | LAC | *Pichia pastoris* | [108] |
| | *Pleurotus sanguineus* | LAC | *Pichia pastoris* | [109] |
| | *Streptomyces cyaneus* | LAC | *Saccharomyces cerevisiae* | [81] |
| | *Gaeumannomyces graminis* | LAC2 | *Cryptococcus* sp. S-2 | [84] |
| | *Coprinus comatus* | LAC3 | *Pichia pastoris* | [110] |
| | *Thermus thermophillus* | LACTt | *Pichia pastoris* | [80] |
| | *Coprinus cinereus* | LCC5I | *Pichia pastoris* | [111] |
| | *Trametes* sp. AH28-2 | LACA | *Saccharomyces cerevisiae* | [82] |
| | *Trametes* sp. AH28-2 | LACB | *Pichia pastoris* | [112] |
| | *Trametes* sp. 420 | LACC | *Pichia pastoris* | [113] |
| | *Trametes* sp. 420 | LACD | *Pichia pastoris* | [114] |
| | *Trametes versicolor* | LCCA | *Pichia pastoris* | [115] |
| | *Trametes versicolor* | LCCB | *Pichia pastoris* | [116] |
| | *Yarrowia lipolytica* | YILAC | *Pichia pastoris* | [72] |
| Lignin peroxidase | *Phanerochaete chrysosporium* BKM-F-1767 | LiPH2 | *Pichia pastoris* | [92] |
| | *Phanerochaete chrysosporium* | LiPH2 | *Saccharomyces cerevisiae* | [95] |
| | *Phanerochaete chrysosporium* | LiPH8 | *Pichia methalonica* | [94] |
| | *Phanerochaete chrysosporium* | LiPH8 | *Saccharomyces cerevisiae* | [117] |
| Manganese-dependent peroxidase | *Phanerochaete chrysosporium* | MnP1 | *Pichia pastoris* | [97] |
| | *Ganoderma lucidum* | MnP1 | *Pichia pastoris* | [101] |
| | *Phanerochaete chrysosporium* | MnP1 | *Saccharomyces cerevisiae* | [102] |
| | *Cerrena unicolor* BB6P | MnP3 | *Pichia pastoris* | [103] |
| Versatile peroxidase | *Pleurotus eryngii* | VPL2 | *Saccharomyces cerevisiae* | [104] |
| | *Pleurotus eryngii* | wtVP | *Saccharomyces cerevisiae* | [118] |

## 4. Other Biochemicals Produced by Yeasts

High tolerance to several lignin-derived compounds may be useful in lignin valorization research [119]. The funneling pathways (metha- and ortho-cleavages) mechanism revealed the potential to use of yeasts to convert lignin-derived compounds into other compounds [120]. In the fermentation of *Trichosporon guehoae*, ferulic acid is reportedly converted into vanillic acid [58]. Another yeast, *T. cutaneum,* showed the ability to convert lignin aldehyde compounds, namely syringaldehyde (2 g/L) and vanillin (2 g/L), into their acids (vanillic acid and syringic acid) and alcohols (syringyl alcohol and vanillyl alcohol) [53]. Various studies have found vanillic and syringic acid to be useful materials in pharmacology, with anti-inflammatory, anti-microbial, and anti-cancer applications [121]. Furthermore, vanillyl alcohol may be of use in the treatment of Parkinson's disease [122].

## 5. Future Perspectives and Feasibility of the Use of Yeasts for Lignin Valorization

Lignin is an abundant biomass material with a complex structure. As a bioresource, it comes directly from nature and is a waste product of the paper, agriculture (pulp), and biorefinery industries. Efforts to maximize its potential have grown significantly through biorefinery research intending to produce desirable compounds. Current innovations in this scope seek to slow the depletion of fossil resources due to their massive consumption in many industrial sectors. Much research has been conducted to investigate the use of lignin as a material for producing lipids, biopolymers, and other valuable aromatic compounds. Compounds derived from lignin can be used in many fields, including energy, food, cosmetics, pharmaceuticals, textiles, and other chemical industries. Nonetheless, most lignin ends up as a low-value material, resulting in further carbon pollution.

Lignocellulosic biomass is mainly composed of carbon, hydrogen, and oxygen, and has been used as a carbon source in the production of biofuels and biochemicals. Additionally, the use of lignin and its derivatives as a carbon source in the natural carbon cycle has gained attention in the last decade. Due to its natural firmness and rigidity, utilizing lignin has always been challenging. Lately, excellent extraction results from biomass are significant for lignin valorization. Several biomass pretreatment methods to overcome the difficulties associated with the utilization of lignin have been developed, such as physical, chemical, and biological. Physical pretreatment methods, including irradiation (microwave and ultrasound), mechanical (milling, grinding, chipping), and heat treatments (hot water treatments), are often used for lignin extractions. However, physical pretreatment has several drawbacks, e.g., high operational cost, high energy use, and the resulting inhibitor byproducts [123]. Meanwhile, chemical pretreatment processes using alkalis (NaOH, Ca(OH)$_2$, and KOH), organic solvents (benzene, hexane, ethanol, and methanol), and strong acids (H$_2$SO$_4$, HCl, and HNO$_3$) have disadvantages of generating pollution and high operational cost [124]. Meanwhile, biological conversion methods offer several advantages regarding their eco-friendly and cost-effective nature. Several microorganisms have the ability to degrade lignin through enzymatic and biochemical processes. Yeast, as a part of the microorganism phylogeny, plays an important role in the biological conversion of lignin. The valorization of lignin by using yeast has been increasing due to its high growth rate, fast genetic manipulation, high tolerance to inhibitors, and relatively easy handling of the cells. The high adaptation level of yeast cells to stress-causing lignin derivates may lead to bioproduct formation in the cells. Several yeasts are capable of converting lignin derivatives into valuable bioproducts such as SCOs, vanillic acid, vanillyl alcohol, and syringic acid. Yeasts are capable of depolymerizing lignin into their monomers and have been employed as a production host for lignin-degrading enzymes from Ascomycota and Basidiomycota fungi. The lignin is broken down by lignin-degrading enzymes (laccases, lignin peroxidases, manganese-dependent peroxidase, and versatile peroxidases) produced by yeasts into lignin derivatives. However, the production is still relatively low and cannot be used on an industrial scale. Hence, in terms of genetic improvements and fermentation steps, optimization should now be the focus. Valorization through the use of yeasts is still limited regarding substrate conversion into desirable products. Hybrid utilization by combining enzyme depolymerization and lignin monomer conversion employing yeasts to produce valuable bioproducts may become an effective strategy to valorize lignin further.

## 6. Conclusions

Lignin and its derivatives are renewable materials with potential value for the production of numerous chemical products, such as lipids, PHAs, vanillin, and high-value acids like vanillic acid and syringic acid. The utilization of lignin resources will reduce the demand for nonrenewable fossil-based fuels and chemicals when those resources are depleted. As a microbial platform to produce valuable products, the valorization of lignin by yeasts might open new avenues of research into lignin valorization.

**Author Contributions:** Conceptualization, F.J.N.P., and P.K.; data curation, F.J.N.P.; visualization, F.J.N.P. and P.K.; writing—original draft, F.J.N.P and P.K; supervision, P.K., A.K., and C.O. All authors have read and agreed to the published version of the manuscript.

**Funding:** This research received no external funding.

**Conflicts of Interest:** The authors declare no conflict of interest.

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
