# Peer review of "Valorization of Lignin and Its Derivatives Using Yeast"

_processes, doi:10.3390/pr10102004_

Round 1
Reviewer 1 Report
There are minor suggestions to improve this manuscript:
1) Figure 2: A piechart is not the best way to represent the composition of various lignocellulosic resources. It is limited to the chosen category (like hardwood or softwood) and the presentation is jumbled. Instead, a table is recommended, where you can pick up to 10 different species and present their classification as well as lignin content. Lignin is also present in fiber crops (not just wood, straw, and grasses) and food waste (e.g. coffee grounds).
2) Spelling and grammatical errors:
1.1 Lignin resources: (suggestion for new subtitle)
1.2. Structure of lignin (not Structural)
3. Value addition of lignin employing yeast
3.2 Enzymes for lignin degradation (this is a subtitle and there is no need to mention yeasts again)
4. Other biochemicals produced by yeasts
3) Muconic acid is mentioned for the first time in Conclusion. This is not advisable. If the three chemicals highlighted in this review are syringic acid, vanillic acid and vanillin, please stick to those and do not introduce new chemicals at the last moment.
4) Figure 3b caption: What is the unit ppu? Please expand. This is not a commonly used symbol.
5) A thorough grammar check is recommended.
Author Response
Response to the comments of Reviewer #1
Comment 1
Figure 2: A pie chart is not the best way to represent the composition of various lignocellulosic resources. It is limited to the chosen category (like hardwood or softwood) and the presentation is jumbled. Instead, a table is recommended, where you can pick up to 10 different species and present their classification as well as lignin content. Lignin is also present in fiber crops (not just wood, straw, and grasses) and food waste (e.g. coffee grounds).
Response comment 1
Thank you for the helpful suggestion to improve the quality of our manuscript. As suggested by the reviewer, we changed “Figure 2” (a pie chart) to “Table 1” (a table style) and present the relative composition of lignocellulosic biomass based on their classification (Page 2).
Comment 2
Spelling and grammatical errors:
1.1 Lignin resources: (suggestion for new subtitle)
1.2. Structure of lignin (not Structural)
- Valueaddition of lignin employing yeast
3.2 Enzymes for lignin degradation (this is a subtitle and there is no need to mention yeasts again)
- Other biochemicalsproduced by yeasts
Response comment 2
Thank you for the comment. We revised the manuscript by changing:
- “1.1. Lignin” to “1.1 Lignin resources” (Line 42).
- “1.2. Structural of lignin” to “1.2. Structure of lignin” (Line 62).
- “3. Value-added product of lignin employing yeast” to “3. Value addition of lignin employing yeast” (Line 100).
- “3.2. The portrayal of enzyme for lignin degradation by yeast” to “3.2. Enzyme for lignin degradation” (Line 148).
- “4. Another biochemical produced by yeasts” to “4. Other biochemicals produced by yeasts” (Line 249)
Comment 3
Muconic acid is mentioned for the first time in Conclusion. This is not advisable. If the three chemicals highlighted in this review are syringic acid, vanillic acid and vanillin, please stick to those and do not introduce new chemicals at the last moment.
Response comment 3
Thank you for the insightful comment. We revised the manuscript by changing “mucoid acid” to “syringic acid and vanillic acid” (Line 314).
Comment 4
Figure 3b caption: What is the unit ppu? Please expand. This is not a commonly used symbol.
Response comment 4
Thank you for the suggestion. We revised the unit “ppu” to “phenylpropane units” (Line 87)
Comment 5
A thorough grammar check is recommended.
Response comment 5
Thank you for the comment. We have done the grammar check.

Reviewer 2 Report
1) Line 52-54: Production of lignin reported annually reaches around 50 million tons from the pulp and paper industry with more than 95% used to produce heat and generate electricity. Ref.?
Recent data indicated around 100 million-tone lignin production as by-product of pulping and bio-refining process. Please add ref.
2) What are figure 1 and 2 ref.?
3) As a review, it is necessary to show results reported by ref. and give a perspective about the published article about the topic. Almost, there is no data about effect of yeast on lignin structure and product results. Authors only reported the name of the articles published about the topic.
4) It is hard to understand what the effect of yeast on lignin is and how can be useful for the lignin vulgarization. there are good number of review out there about the topic with more details. Need to focus on the results reported by the researchers so far and give a perspective about that.
Author Response
Response to the comments of Reviewer #2
Comment 1
Line 52-54: Production of lignin reported annually reaches around 50 million tons from the pulp and paper industry with more than 95% used to produce heat and generate electricity. Ref.?
Recent data indicated around 100 million-tone lignin production as by-product of pulping and bio-refining process. Please add ref.
Response comment 1
Thank you for the insightful suggestion to help improve the quality of our manuscript. As suggested by the reviewer, we updated the data with a recent reported article from Bajwa et al., 2019 entitled A Concise Review of Current Lignin Production, Applications, Products and Their Environmental Impact. Ind Crops Prod. We revised the word “Production of lignin reported annually reaches around 50 million tons from the pulp and paper industry with more than 95% used to produce heat and generate electricity.” to “Global production of lignin reported annually reaches around 100 million tons from the paper industry and ethanol waste production” (Line 54-55). We also added the word “Most of the lignin waste is reported used as a low-value substrate to produce heat and generate electricity” (Line 55-56) based on the article reported by Grossman & Vermerris, 2019 entitled Lignin-Based Polymers and Nanomaterials. Curr Opin Biotechnol.
Comment 2
What are figure 1 and 2 ref.?
Response comment 2
Thank you for the comment. In the previous manuscript, we put Figure 1 in the manuscript as a graphical abstract. However, it was not a proper place. Therefore, we removed the previous “Figure 1” and moved it to a graphical abstract section and changed the previous “Figure 3” to “Figure 1” in the manuscript. Reference of “Figure 2” was written in line 43 as “[9]” reference from an article reported by Sorieul et al., 2016 entitled Plant Fibre: Molecular Structure and Biomechanical Properties, of a Complex Living Material, Influencing Its Deconstruction towards a Biobased Composite. Materials (in the previous manuscript). As suggested by reviewer 1, we changed the pie chart style of “Figure 2” to “Table 1. Relative composition of lignocellulosic biomass” (table style) for better data presentation.
Comment 3
As a review, it is necessary to show results reported by ref. and give a perspective about the published article about the topic. Almost, there is no data about effect of yeast on lignin structure and product results. Authors only reported the name of the articles published about the topic.
Response comment 3
Thank you for the helpful comment to improve our manuscript. As suggested by the reviewer, we added more data results in the manuscript as follows:
- Subsection 3.1 in lines 118 and 119 related to the lignin aldehydes concentration tolerance of Trichosporon cutaneum. In line 121 related to lipid production of cutaneum. Lines 129 and 130 related to the concentration tolerance, line 136 related to the lipid production of T. oleaginosus and added some text related to the specific mechanisms of lipid production from lignin aldehyde (4-Hydroxybenzaldehyde) using reference. “4-Hydroxybenzaldehyde is converted into 4-hydroxybenzoate by aldehyde dehydrogenases of T. cutaneum. Afterward, it is assimilated into protocatechuate by dioxygenases with oxygen as the substrate. Protocatechuate is converted into acetyl-CoA and succinyl-CoA via β-ketoadipate pathway. The acetyl-CoA is used for lipid synthesis directly or assimilated first into TCA cycle and used as the dominant precursor for lipid synthesis” (Line 123-128).
- Subsection 3.2 in line 179 related to the laccase production of Rhodotorula mucilaginosa. Lines 220-222 related to peroxidase production. We added some text related to the laccase production results of Kluyveromyces dobzhanskii DW1 and Pichia manshurica DW2 using reference. “Rice bran substrate supported the laccase production of 88.625 U/L in dobzhanskii DW1 and sugarcane bagasse of 79.107 U/L in P. manshurica DW2” (Line 181-183).
- Section 4 in line 257 we added results related to syringaldehyde and vanillin concentrations.
Comment 4
It is hard to understand what the effect of yeast on lignin is and how can be useful for the lignin vulgarization. there are good number of review out there about the topic with more details. Need to focus on the results reported by the researchers so far and give a perspective about that.
Response comment 4
Thank you for the helpful comment. We added results of lipid production in lines 121 and 136. We added some text related to the mechanism of lipid production from lignin aldehyde (4-Hydroxybenzaldehyde) using reference. “4-Hydroxybenzaldehyde is converted into 4-hydroxybenzoate by aldehyde dehydrogenases of T. cutaneum. Afterward, it is assimilated into protocatechuate by dioxygenases with oxygen as the substrate. Protocatechuate is converted into acetyl-CoA and succinyl-CoA via β-ketoadipate pathway. The acetyl-CoA is used for lipid synthesis directly or assimilated first into TCA cycle and used as the dominant precursor for lipid synthesis” (Line 123-128). We described the perspective in section 5 related to the valorization of yeast to produce the valuable product from lignin derivates. “The valorization of lignin by using yeast has been increasing significantly due to its merits including high growth rate, fast genetic manipulation, high tolerance to inhibitors, and relatively easy handling of the cells. The high adaptation level of yeast cells to stress-causing lignin derivates presence may lead to producing bioproduct formation in the cells. Several yeasts are capable to utilize lignin derivatives into other valuable bioproducts such as SCOs, vanillic acid, vanillyl alcohol, and syringic acid.” (Line 293-299). We also added some text related to the importance of enzymes as a part of lignin valorization. “The enzymes are important to catalyze the lignin into its derivatives or aromatic monomers which provide for building blocks of some valuable chemical products” (Line 150-152). We also added some text related to the laccase production results of Kluyveromyces dobzhanskii DW1 and Pichia manshurica DW2 using reference. “Rice bran substrate supported the laccase production of 88.625 U/L in K. dobzhanskii DW1 and sugarcane bagasse of 79.107 U/L in P. manshurica DW2” (Line 181-183). We described the perspective of the lignin-degrading enzyme in section 5. “The yeasts are capable to depolymerize lignin into its monomers and utilize into valuable compounds. Yeast cells have been employed as a production host for lignin-degrading enzymes from Ascomycota and Basidiomycota fungi. The lignin will be breakdown by the lignin-degrading enzymes (laccases, lignin peroxidases, manganese-dependent peroxidase, and versatile peroxidases) produced by yeasts into lignin derivatives. However, the production is still relatively low and cannot be used on an industrial scale yet. Hence, optimization regarding genetic improvement and fermentation steps should be the next focus for increasing the production titer. Currently, valorization by employing yeasts is still in the narrow range regarding substrate conversion into the products. Hybrid utilization by the combination of the enzyme depolymerization and lignin monomer conversion employing yeasts to produce valuable bioproducts may become an effective strategy to achieve better lignin valorization” (Line 299-310).

Round 2
Reviewer 2 Report
n/a